# Learning Robotic Locomotion Affordances and Photorealistic Simulators from Human-Captured Data

**Alejandro Escontrela**     **Justin Kerr**     **Kyle Stachowicz**     **Pieter Abbeel**

University of California, Berkeley

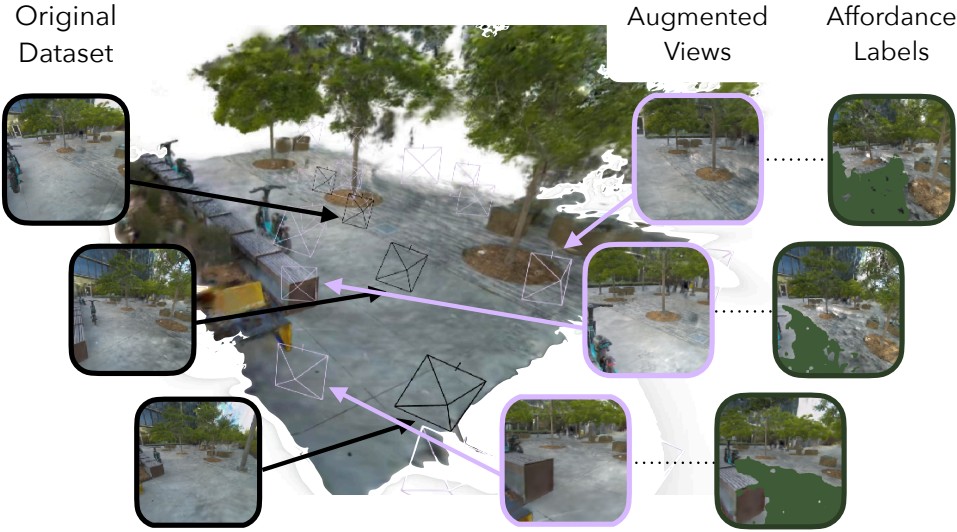

Figure 1: PAWS expands the affordance model's training data well beyond the original images by using the learned radiance field to generate novel views of the scene. Affordance labels are generated by comparing DINOv2 features near foot poses to other parts of the scene.

**Abstract:** Learning reliable affordance models which satisfy human preferences is often hindered by lack of high-quality training data. Similarly, learning visuomotor policies in simulation can be challenging due to the high cost of photo-realistic rendering. We present PAWS: a comprehensive robot learning framework that uses a novel portable data capture rig and processing pipeline to collect long-horizon trajectories that include camera poses, foot poses, terrain meshes, and 3D radiance fields. We also contribute PAWS-Data: an extensive dataset gathered with PAWS containing over 10 hours of indoor and outdoor trajectories spanning a variety of scenes. With PAWS-Data we leverage radiance fields' photo-realistic rendering to generate tens of thousands of viewpoint-augmented images, then produce pixel affordance labels by identifying semantically similar regions to those traversed by the user. On this data we finetune a navigation affordance model from a pretrained backbone, and perform detailed ablations. Additionally, We open source PAWS-Sim, a high-speed photo-realistic simulator which integrates PAWS-Data with IsaacSim, enabling research for visuomotor policy learning. We evaluate the utility of the affordance model on a quadrupedal robot, which plans through affordances to follow pathways and sidewalks, and avoid human collisions. Project resources are available on the website.

**Keywords:** Navigation, Dataset, Real2Sim

8th Conference on Robot Learning (CoRL 2024), Munich, Germany.

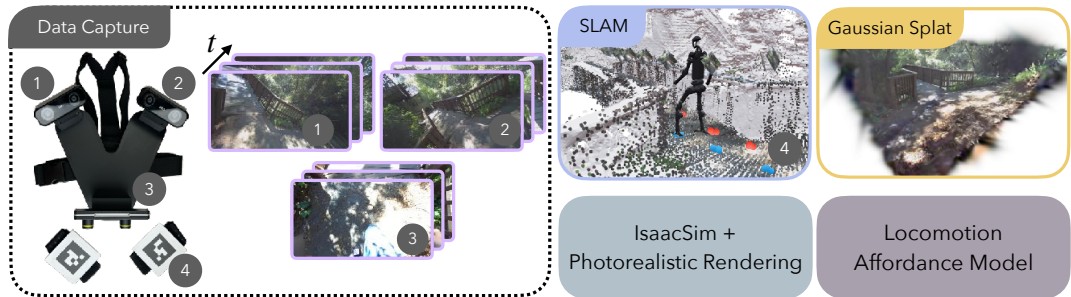

Figure 2: **PAWS** proposes a data capture pipeline consisting of three cameras positioned on the shoulders (1, 2) and facing down (3), as well as a fiducial marker attached to each foot (4). RGB and depth streams are processed with SLAM to estimate camera poses while fiducial markers provide foot poses. The processed data is then used to extract terrain meshes and train 3D Gaussian radiance fields with Splatfacto. We utilize the processed dataset to train a powerful locomotion affordance model and develop a fast and photo-realistic simulator.

## 1 Introduction

Ensuring that robots navigate in the wild while satisfying human preferences is crucial to ensure safe deployment. Without considering navigation affordances, our robots may perilously ignore cross-walks, stray off of sidewalks, trespass on forbidden ground, or damage themselves on hazards. However, current algorithms for learning affordances and training visuomotor policies often suffer from a lack of high-quality training data. Additionally, defining effective affordance labels from human data is challenging, as the naive approach of assigning high affordances solely to traversed areas leads to sparse labels with many false negatives. Further, simulators for training visuomotor policies tend to provide low-fidelity renders.

In this work we propose PAWS, a method for instrumenting a person with cameras to capture ego view of natural human locomotion in the wild for affordance model training. PAWS can record long-horizon trajectories of indoor or outdoor scenes and process them to obtain camera poses, foot positions, radiance field reconstructions, and terrain meshes. Using this method we contribute PAWS-Data, a dataset of over ten hours of the aforementioned data spanning diverse scenes including hikes, cityscapes, and indoor environments. We also integrate PAWS-Data with IsaacSim (Mittal et al., 2023) to develop a fast and realistic simulator which can generate thousands of photo-realistic images per second across thousands of environments.

Neural Radiance Fields have exploded in popularity as a photo-realistic reconstruction method (Mildenhall et al., 2020), however they are prohibitively expensive to train and render at scale. Recently, Gaussian Splatting (Kerbl et al., 2023) significantly reduces the cost of rendering by eliminating the need for expensive ray casting operations, opening the door for real-time photo-realistic reconstruction in robot data pipelines. We take advantage of this fact to reconstruct photo-realistically renderable environments from human locomotion data.

We utilize the PAWS-Data radiance fields to train a Locomotion Affordance Model (PAWS-LAM) on a large augmented dataset by rendering from novel views while retaining visual fidelity. Affordance labels are generated using DINO features to label parts of the scene which are semantically similar to areas traversed by the user. In our experiments, we demonstrate that viewpoint augmentation significantly enhances the prediction accuracy for trained models, while semantic-aware affordance labels accurately capture human preferences for navigation. We also find that a foundation model backbone enables generalization to different environments beyond the training data. We apply PAWS-LAM to a quadruped navigation task, and combine PAWS-LAM with a simple planner to produce robot trajectories which are safe and efficient.

## 2 Approach

In this section we discuss the three contributions that make up our method: PAWS: a data collection system for capturing and processing a large quantity of trajectories, PAWS-LAM: A robotic locomotion affordance model that leverages learned radiance fields and semantic features to produce high

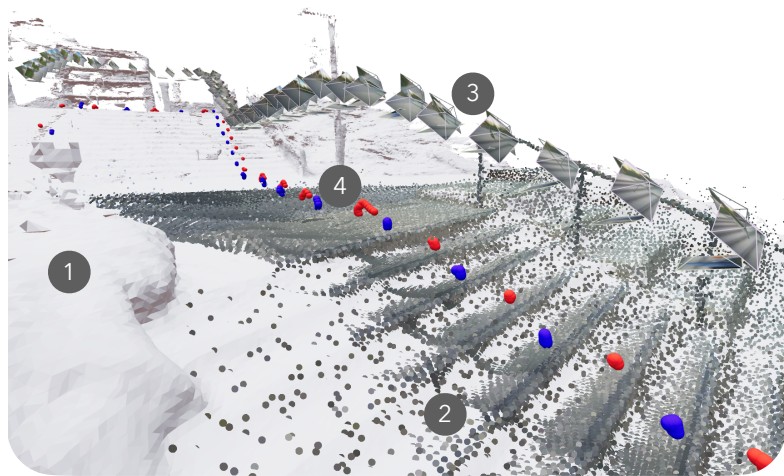

Figure 3: PAWS-Data provides many data products derived from the original RGB, Depth, and IMU streams, including scene meshes (1), pointclouds (2), camera poses (3), and foot positions (4).

quality affordance predictions, and PAWS-Sim: which leverages the PAWS-Data dataset and a massively parallel simulators to achieve fast photo-realistic rendering across thousands of environments.

## 2.1 Data Collection

PAWS uses three cameras mounted on the user's torso to capture three stereo camera streams. Two cameras are mounted on the shoulder and angled such that they capture a wide field of view while attending to the terrain ahead. A third camera is pointed directly down towards the user's feet, and is used for two functions: to track the fiducial markers placed on the user's shoes, and to provide a clear top down view of the terrain for later use when training 3D Gaussian radiance fields. Figure 2 demonstrates an overview of the camera setup and our data processing steps. We use DROID-SLAM (Teed and Deng, 2021) to provide an estimate of the camera poses and to generate a colored point cloud of the environment. Importantly, we find that initializing DROID-SLAM disparities with stereo depth values significantly improves the quality of estimated poses. We also compute human segmentation masks with Mask2Former (Cheng et al., 2022) to mask out humans for later processing steps. Given the estimated camera poses, we can estimate foot poses in the global frame using the relative poses provided by the fiducial markers (Olson, 2011). Re-projecting depth from the known camera poses allows us to generate an accurate mesh of the environment, capturing details such as stairs and rough terrains. Finally, the RGB, Depth, camera poses, and point cloud are used to train a 3D Gaussian splatting model of the scene using Splatfacto (Ye et al., 2023; Tancik et al., 2023). Figure 3 visualizes the data products produced by PAWS. PAWS-Data consists of 10 hours of diverse indoor and outdoor trajectories, a subset of which are visualized in Figure 6.

## 2.2 Locomotion Affordance Model

**Dataset Augmentation** Given a dataset consisting of radiance fields and camera poses, we significantly expand the dataset size by rendering from novel viewpoints, a process we refer to as Viewpoint Augmentation. Namely, we sample a batch of new camera poses by perturbing the positions and orientations of the original camera poses by up to 3 meters and 30 degrees, respectively. We then rasterize images from the sampled camera poses using the learned 3D Gaussian radiance fields, which allow for fast batch rendering. Figure 1 visualizes a radiance field along with the original camera poses, augmented poses, and affordance labels defined in the next section.

**Affordance Label Generation** Given a dataset of camera images, camera poses, and foot positions, a naive approach to generate labels would be to simply project the foot position into the camera image. While these labels accurately capture where the human navigated, they are sparse and falsely label possibly navigable areas as having low affordance. Instead, we propose to label regions which are

Gaussian Splatting Render      IsaacSim      RGB     Depth

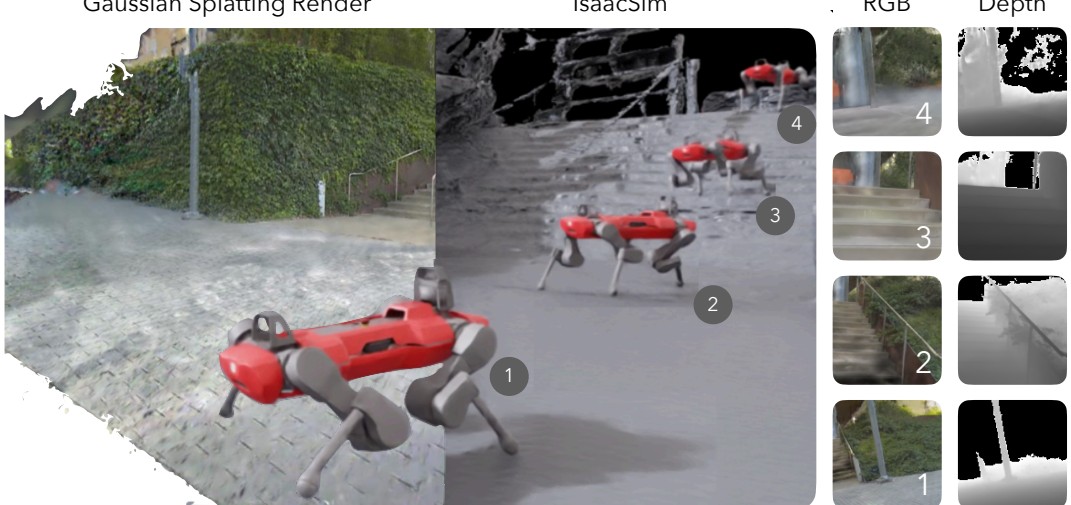

Figure 4: **PAWS-Sim** is a fast, photo-realistic environment suite which leverages the 3D Gaussian radiance fields, terrain meshes, and poses from PAWS-Data to generate thousands of high-fidelity renders per second.

*semantically similar* to those traversed by the human. We do so by computing DINOv2 features (Caron et al., 2021) for the $M$ patches the human stepped on in the scene. We then compute a $M \times N \times N$ cosine similarity matrix, comparing all $M$ traversed ground features to the $N \times N$ features in the input image. We take the minimum along the footstep feature dimension and threshold by a $c_{\text{DINO}}$ to get affordance patches, and upsample with bi-linear sampling to obtain affordance labels. Figure 5 compares the naive dilated sparse affordance labels to the affordance labels produced by PAWS.

**Model Architecture** Given the affordance dataset from the previous section, we fine-tune a foundation model backbone on the task of affordance prediction. Specifically, we attach a simple token projection head to a pre-trained DINOv2 (Caron et al., 2021) model and train the model to minimize the cross entropy loss against the affordance labels. In Section 3.1 we evaluate different choices of backbone and model size.

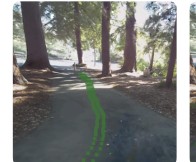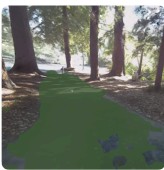

Figure 5: Naive v.s. PAWS affordance labels

### 2.3 Fast and Photo-realistic Simulation

PAWS-Data provides camera poses, foot poses, meshes, and radiance fields that accurately match the scale of objects in the real world. Additionally, 3D Gaussian radiance fields allow for fast rendering, allowing for thousands of renders per seconds. We integrate these features into IsaacSim (Mittal et al., 2023) to develop an open source photo-realistic gym environment for running robots which can be adapted to any desired reward signal or user-captured scene. Namely, we create a convex decomposition of the 3D terrain mesh to allow for stable physics simulation. In `env.step`, we query the 3D Gaussians for each scene with a batch containing each robot's camera pose, camera intrinsics, and image resolution to obtain thousands of renders per second. While 3D Gaussian splats provide high-fidelity renders for a large portion of poses within the scene, deviating significantly from the original camera poses can lead to highly distorted and unrealistic renders. To combat this, we initialize the robots near the original camera poses, and terminate the episode once the camera pose has deviated significantly from the set of original camera poses, while still bootstrapping values.

## 3 Experiments

In this section we evaluate PAWS-LAM and assess it's utility in a practical robotics setting. Specifically, we aim to answer the following research questions:

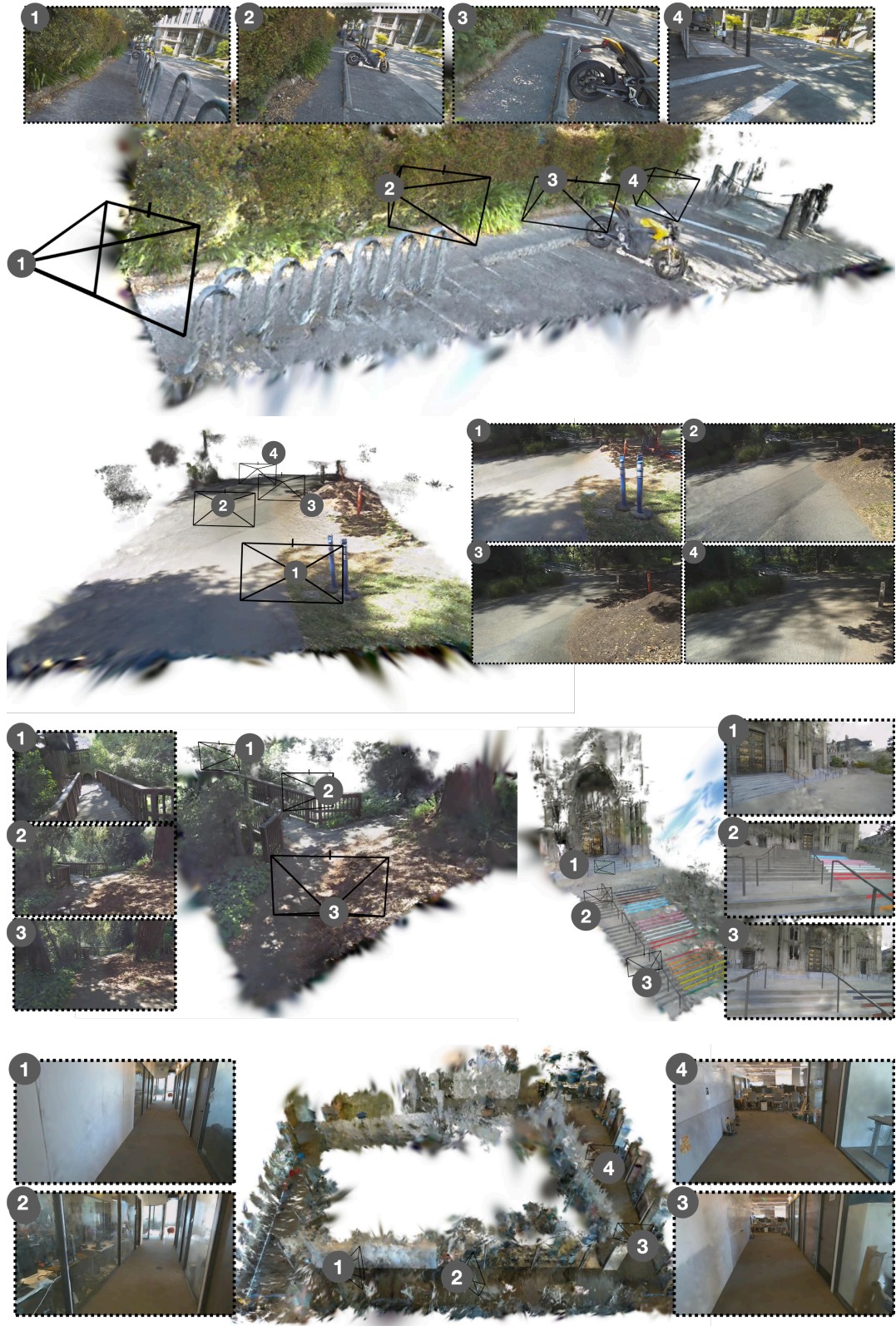

Figure 6: Example scenes in PAWS-Data. Aside from radiance fields, the PAWS dataset also contains terrain meshes, foot positions, and affordance labels. Artifacts in aerial renders can be attributed to the extreme viewing angle.

- How does PAWS-LAM compare to alternative approaches to affordance learning?
- How well does PAWS-LAM generalize to unseen environments?
- Does PAWS-LAM learn affordances that facilitate safe quadruped navigation?

## 3.1 Locomotion Affordance Model Evaluation

We ablate different design decisions, including training with viewpoint augmentation and different backbone sizes. In Table Figure 7b, we ablate PAWS-LAM trained with and without viewpoint augmentation. We find that disabling viewpoint augmentation drastically reduces performance on the validation set, as measured by pixel accuracy and mean intersection over union (mIoU). We attribute this to the decreased dataset size, as well as a smaller set of distinct camera poses, which both contribute to overfitting. Figure 7a visually demonstrates the failure modes of training without viewpoint augmentation. Specifically, models trained without viewpoint augmentation fail to capture the nuanced human preferences encountered in complex scenarios, such as predicting the entire road as navigable while only sidewalks or crosswalks should be traversed. Additionally, increasing the size of the pre-trained backbone from 21M parameters (ViT-S/14) to 86M parameters (ViT-B/14) significantly enhances model performance.

Additionally, in Table Figure 7c we evaluate PAWS-LAM's ability to generalize to environments unseen during training. We evaluate a model trained solely on outdoor data on a held-out dataset of indoor trajectories and find that PAWS-LAM generalizes favorably, displaying a slight decrease in performance. Further, models trained without viewpoint augmentation generalize poorly to the indoor dataset, which we hypothesize is due to the model overfitting to a small set of DINOV2 features which do not generalize to indoor environments.

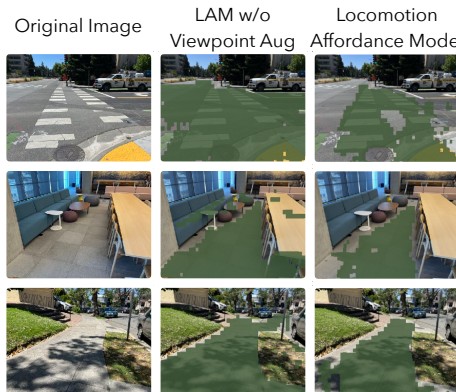

(a) Locomotion Affordance Model (LAM) predictions on exemplary held-out test images.

| Method | mIoU | Pix Acc |
|---|---|---|
| LAM w/o View Aug | $0.61 \pm 2.6$ | $0.72 \pm 2.9$ |
| LAM (ViT-S/14) | $0.76 \pm 1.5$ | $0.84 \pm 1.5$ |
| LAM (ViT-B/14) | $\mathbf{0.83 \pm 1.8}$ | $\mathbf{0.91 \pm 1.7}$ |

(b) Locomotion affordance model ablations.

| Method | mIoU | Pix Acc |
|---|---|---|
| LAM w/o View Aug | $0.38 \pm 1.2$ | $0.44 \pm 1.4$ |
| LAM (ViT-S/14) | $0.69 \pm 2.3$ | $75 \pm 2.1$ |
| LAM (ViT-B/14) | $\mathbf{0.73 \pm 1.9}$ | $\mathbf{0.81 \pm 2.1}$ |

(c) Locomotion affordance model generalization performance on held-out indoor dataset. Model was only trained on outdoor data.

Figure 7: Locomotion Affordance Models trained with viewpoint augmentation more accurately capture human navigation preferences such as staying on crosswalks and sidewalks and not hitting obstacles.

## 3.2 Robotic Navigation

We now propose one potential downstream application of our data collection interface and splat-based rendering pipeline: learning affordance models that transfer across embodiments, from human data to robot locomotion. We propose a simple pipeline using our affordance model to optimize a robot's high-level plan to ensure that it is physically reasonable and matches human affordances such as sticking to marked paths or avoiding other humans.

We use a Unitree GO1 quadruped as our target platform, equipped with two depth cameras to capture a wide field of view of the scene. For each camera, we compute visual affordances using PAWS-LAM, then project the predicted affordance probabilities directly onto point clouds from the depth cameras. This map is then flattened and used as a 2-dimensional cost map, through which any planner can be

used to to generate a high-level plan. For simplicity, we use a grid-based Dijkstra's algorithm, but the generally proposed method is applicable to any sufficiently flexible planning mechanism. Figure 8a visualizes an example plan through the affordance map.

As shown in Figure 8, we find that the resulting system is able to match human affordances in an intuitive manner, sticking to paths whenever possible and following forks in the road. Importantly, the robot's camera perspective is *not* contained in the initial dataset. Suggesting that the viewpoint augmentation used for PAWS-LAM helps to obtain high downstream performance on a non-humanoid embodiment. Please see our supplementary materials for a reference implementation of the planning algorithm and videos of our robot following human-interpretable affordances.

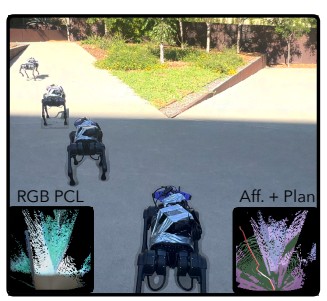 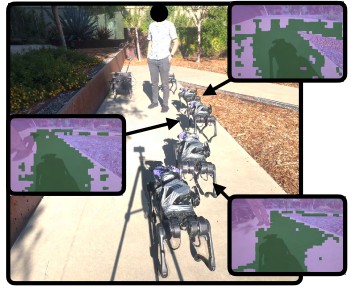 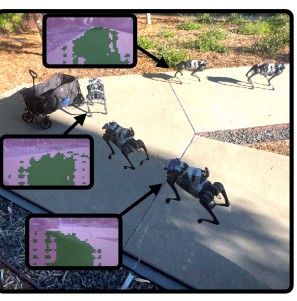

(a) Fork in the road        (b) Human avoidance        (c) Turn

Figure 8: Planning through learned affordances enables the quadrupedal robot to navigate safely without violating human preferences over navigation.

## 4 Related Work

### 4.1 Large Scale Photo-realistic Reconstruction

Scene reconstruction has long been studied, classically in the context of SLAM or SfM reconstruction (Newcombe et al., 2011; Schönberger and Frahm, 2016; Mur-Artal et al., 2015). More recently, advances in neural rendering have drastically improved the capabilities of these systems for rendering visual outputs. Neural Radiance Fields (NeRF) (Mildenhall et al., 2020) can render highly photo-realistic images from novel views, and an explosion of extensions expand on the speed (Müller et al., 2022), quality (Barron et al., 2021; 2022; 2023), scene scale (Tancik et al., 2022), and tooling (Tancik et al., 2023). Recently, 3D Gaussian Splatting (3DGS) (Kerbl et al., 2023) proposes a method of representing a scene with a collection of oriented 3D Gaussians, which can be rasterized onto an image quickly with modern GPU hardware. This work uses 3DGS to reconstruct the surroundings of human capture trajectories for fast rendering at volume.

### 4.2 Learning Affordances for Robot Locomotion

For mobile robots to efficiently navigate the world while respecting both the laws of physics and social rules, it is important that they have some notion of *affordances* describing which states they can enter and which actions they can perform. Traditional legged locomotion policy learning architectures learn these affordances in simulation with domain randomization (Feng et al., 2023; Lee et al., 2020; Margolis et al., 2023) relying on either blind models (Kumar et al., 2021) or depth information (Agarwal et al., 2023). However, many more subtle cues cannot in fact be recovered by depth information alone: a concrete road and a sand pit might appear identical when viewed as a depth image (Loquercio et al., 2023), and they have difficulty resolving transparent or translucent objects. Furthermore, using RGB cameras instead of depth can allow *socially-compliant* behavior rather than purely physical behavior: walking on paths and in crosswalks rather than through a grass field, for example. Terrestrial navigation research has also studied the problem of traversability estimation from visual data, particularly in the off-road driving case (Jung et al., 2024). In contrast to these works, we do not require hand-tuned heuristics nor are our learned affordances limited to only physical parameters:

PAWS can learn rich, interpretable human affordances from features in in both dense urban environments and outdoor settings, and can be used to provide a dense traversability score rather than just a 0-1 traversability map if desired.

### 4.3 Photo-realistic Simulators

Developing high-fidelity simulators is a large and active area of research, spanning applications in autonomous driving (Amini et al., 2021; Dosovitskiy et al., 2017), UAV research (Shah et al., 2017; Guerra et al., 2019), and robotics (Mittal et al., 2023; Gan et al., 2020). While these simulators can achieve high-quality renders, they often require high compute budgets which decreases sample throughput. Additionally, designing simulation can be time consuming, and even the highest quality simulators may not fully overcome the simulation to reality (Sim2Real) gap. These challenges have spurred researchers to find new solutions to the challenges of overcoming the Sim2Real gap. Bousmalis et al. and Shrivastava et al. explored pixel-level domain adaptation by training networks to post-process the simulated render to appear more realistic. While exciting, these approaches are difficult to train due to a lack of paired training data and can suffer from unwanted hallucinations. Tirumala et al. leverage Neural Radiance Fields (NeRF) to generate photo-realistic renders for an indoor robot soccer environment. While promising, rendering with NeRFs requires expensive ray tracing operations, which can be prohibitively slow when compared to rasterization, and NeRFs to not allow for efficient batch rendering. Additionally, they limited their focus to one particular environment. In contrast, PAWS-Sim utilizes 3DGS as a rendering backbone, which enable batch rendering at speeds that are multiple orders of magnitude higher, while retaining high visual fidelity. Additionally, PAWS-Sim support the over 10 hours of scenes present in PAWS-Data, spanning a wide variety of environments.

## 5 Discussion

We present PAWS, a self-supervised data collection method for obtaining large datasets of scenes from human captured data. Accompanying PAWS is PAWS-Data, a dataset of over 10 hours of indoor and outdoor trajectories gathered with PAWS, containing RGB + Depth, camera poses, foot poses, terrain meshes, and radiance fields. We also introduces PAWS-LAM, which leverages PAWS-Data to train a locomotion affordance model capable of modeling the nuanced human preferences of outdoor navigation. We ablate PAWS-LAM and show that it benefits from our novel affordance dataset augmentation approach, our proposed affordance label generation method which drastically reduces false negatives, and the use of a foundation model backbone. Additionally, PAWS-LAM generalizes well to unseen environments, which we showcase by evaluating on indoor scenes using a model trained solely on outdoor data. We show how PAWS-LAM can be used for robotic navigation by using the affordance model to optimize a high-level plan to ensure the robot's trajectory satisfies human navigation preferences. PAWS-LAM generalizes to the robot's distribution of observed images, showcasing favorable generalization across embodiments. Finally, we introduce PAWS-Sim, a fast and photo-realistic simulator which integrates PAWS-Data IsaacSim to enable fast visuomotor policy learning.

**Limitations & Future Work** While PAWS can gather complete trajectories for the camera pose, the foot goes out of frame for roughly half the user's gait, thereby limiting foot position estimation for that duration. Additionally, fast motions can produce motion blur, making it difficult to detect the fiducial marker. Approaches such as SelfPose (Tomè et al., 2020) estimate body positions from a headmounted camera and a pose detection network. Such approaches would potentially increase foot position availability and decrease hardware complexity. Further, integrating terrain dynamics estimation with our method would increase the physical fidelity of the simulator, further reducing the Sim2Real gap.

**Acknowledgments**

We thank Philipp Wu for his assistance in the design and manufacture of the data capture rig. We also thank Oleh Rybkin and Carmelo Sferrazza for providing insightful feedback that helped guide the project. This work was supported in part by an NSF Fellowship, NSF GRF #2146752. Pieter Abbeel holds concurrent appointments as a Professor at UC Berkeley and as an Amazon Scholar. This paper describes work performed at UC Berkeley and is not associated with Amazon.

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

## A  Gaussian Splatting Details

We initialize the 3DGS start points with points deprojected from DROID-SLAM disparity, de-duplicated. To accomodate small amounts of SLAM drift, we enable camera optimization, similar to (Seiskari et al., 2024; Zhao et al., 2024), which refines poses during 3DGS training. To deal with the large scene scale in our captures, we need to modify a few important hyperparameters inside Splatfacto, which was primarily tuned for use with object-centric captures. We use the AbsGS splitting heuristic introduced by (Ye et al., 2024) with a gradient threshold of $0.0006$ as well as lowered learning rate on the Adam optimizer controlling 3DGS means to $5e - 5$ as recommended by the original implementation. We also extend the length of the culling/splitting portion of 3DGS training from 15000 to 25000 steps, and lower the densify size threshold to $0.1\%$ from $1\%$, ensuring fine details are reconstructed.

## B  Hardware Specifications

The PAWS data collection system consists of three cameras, one on each shoulder, and a third camera pointed at the feet. The shoulder-mounted cameras are stereo ZedX Mini cameras, while the downward facing camera is a stereo Zed Mini. For depth estimation we opt for the provided neural depth model, which is slightly less accurate than RAFT-Stereo (Lipson et al., 2021) yet considerably faster. The three cameras are connected to a rigid plate attached to the chest, and their precise transforms are derived from the CAD model. All three cameras are connected to a NVIDIA Jetson AGX Orin Developer Kit which hardware encodes and saves the three camera streams for later processing.

## C  Rendering Performance Analysis

3D Gaussian Splatting models support fast batch rendering, capable of rendering thousands of frames per second at certain resolutions.

| Resolution | FPS (RGB + Depth) |
|---|---|
| $64 \times 64$ | $1487 \pm 12$ |
| $96 \times 96$ | $1023 \pm 9$ |
| $256 \times 256$ | $756 \pm 9$ |
| $512 \times 512$ | $431 \pm 6$ |
| $1024 \times 1024$ | $185 \pm 3$ |

Figure C.1: Rendering frames per second for various resolutions. Reported times are for computing both RGB and Depth values from the 3DGS.

