# OpenReview forum: "Learning Robotic Locomotion Affordances and Photorealistic Simulators from Human-Captured Data"
_robot-learning.org/CoRL/2024/Conference — CoRL 2024_

### Official Review · Reviewer_V5mu · 2024-07-16
**Good data collection ideas, but limited technical contributions**

**Originality:** 3
**Technical Quality:** 3
**Clarity Of Presentation:** 5
**Potential Impact:** 3
**Recommendation:** 2
**Confidence:** 3

**Review:**

Strengths:
- Innovative Data Collection: The use of human-captured data to train robotic models ensures that the locomotion policies are aligned with human navigation preferences.
- Viewpoint Augmentation: This method significantly expands the dataset size and diversity, enhancing the robustness and generalization of the affordance model.
- High-Fidelity Simulation: PAWS-Sim provides a realistic simulation environment, crucial for developing and testing visuomotor policies efficiently.
- Semantic Affordance Labeling: Improves the accuracy of affordance models by focusing on semantic similarities, leading to better generalization and fewer false negatives.
- Comprehensive Dataset: PAWS-Data includes diverse environments, offering a rich resource for training and evaluation.

Weaknesses:
- Hardware Complexity: The data capture setup, which includes multiple cameras and fiducial markers, seems cumbersome and may be susceptible to motion blur during rapid movements.
- Simulation Constraints: The simulation environment may not adequately reflect the physical dynamics of real-world terrains, potentially limiting the applicability of the findings to real-world scenarios.
- Limited Technical Contributions: The paper predominantly utilizes existing methods, such as Gaussian Splatting, IsaacSIM, and DINOv2 to train an affordance model. There appears to be a lack of novel technical contributions.

While the ideas presented in the paper are interesting, there are multiple gaps that have not been addressed well --- the difference between the dynamics of a human and a robot are not considered. The most critical point is how can affordances be converted to a specific score associated with a particular robot/planning algorithm (what is traversable for a robot may not be traversable for a human and vice versa). The individual components used in the method are not novel --- projection of affordance labels using DINO features has been done in previous works, collecting data from people's footsteps and associating it with quadruped foothold traversibility has been researched before. The proposed method's main contribution seems to lie in combining such methods together. It would be helpful if the authors could clarify the contributions more clearly. based on the above, the potential impact of the work is also limited.

For limitations, the authors have mentioned the reliance of fiducials and the lack of terrain dynamics. But there is a lack of limitations discussion regarding the method itself (such as how the affordance is used for planning). Previous works address affordance as a continuous score, while this work treats it as a binary score. Further, the method assumes that the robots that would make use of it would have similar affordances as the human -- i.e. ANYmal, Go1. it would be helpful to either address this as a limitation of the method, or specify it as the initial scope within the problem setup. Also, it would be suitable to mention the limitation of creating an elaborate rig for this data collection process as compared to the alternative of simply using a phone or readily available sensors that do not require a sophisticated hardware setup.

In terms of comparisons, it would have been beneficial to compare also with other affordance models, such as "Fast Traversibility Estimation for Wild Visual Navigation" by Frey et al.

**Quality Of The Limitations Section:**

2

**Questions For Rebuttal:**

1. Please provide a detailed explanation of the process used to convert the "affordance" (the output of the model) into a score for planning.
2. Could you elaborate on how motion blur affects the quality of reconstruction? Did you explicitly select keyframes to mitigate this issue?
3. The assumption underlying the data collection method requires further clarification, especially considering the differences in dynamics between humans and legged robots.

**Robotics Focus:**

4

**Summary Of Paper:**

The paper presents PAWS, a comprehensive robot learning framework designed to enhance robot navigation by leveraging human-captured data. The framework encompasses three main contributions: (1) PAWS Data Collection System: A portable data capture rig and processing pipeline for collecting long-horizon trajectories; (2) PAWS-Sim: Using Gaussian Splatting to construct a high-speed, photo-realistic simulation in IsaacSim using the PAWS-Data; (3) PAWS-LAM: A robot locomotion affordance model, which is fine-tuned on a dataset enriched through novel viewpoint augmentation with the DINOv2 model and viewpoint projection.

**Summary Of Recommendation:**

I recommend Weak Reject because of the lack of technical novelty in the method, however I am willing to change my recommendation either way depending on the author responses and reviewer comments

---

### Official Review · Reviewer_gXMo · 2024-07-18
**A pipeline for collecting real-world data, affordances, and usage in affordance prediction and downstream tasks**

**Originality:** 3
**Technical Quality:** 4
**Clarity Of Presentation:** 4
**Potential Impact:** 3
**Recommendation:** 3
**Confidence:** 3

**Review:**

The pipeline proposed in this paper is a complete procedure for collecting rich sensory data of natural scenes. Along with the data collection pipeline, the paper also proposes a photo-realistic simulator and affordance prediction models.

The experiments validate the effectiveness of having 3D scene reconstructions over 2D affordances, as well as the effectiveness of a larger backbone architecture. The on-robot study shows that learned affordance can be used in navigation tasks.

Overall, the paper is clearly written and a solid contribution to the CoRL community. The completeness of the proposed pipeline is much appreciated. The key design choice of having 3D reconstruction is validated in experiments.

The on-robot validation may bear some room for improvement: for example, in more diverse scenarios/scenes.

Another related work to this paper: Margolis, Gabriel B., et al. "Learning to see physical properties with active sensing motor policies." arXiv preprint arXiv:2311.01405 (2023).

**Quality Of The Limitations Section:**

3

**Questions For Rebuttal:**

The photo-realistic simulator is proposed but it is unclear how useful it might be for downstream tasks such as policy learning. It would be great (but not necessary, in this reviewer's opinion) to have experiments to validate if the simulator could be used for a downstream task.

**Robotics Focus:**

4

**Summary Of Paper:**

This paper proposes a pipeline for collecting human-captured real-world data and affordances. Such data can be used for affordance prediction and building a photo-realistic simulator. The learned affordances can be used for guiding a navigation policy.

**Summary Of Recommendation:**

Overall a good paper -- experimental validation could be further improved.

---

### Official Review · Reviewer_ZQSK · 2024-07-23

**Originality:** 4
**Technical Quality:** 4
**Clarity Of Presentation:** 4
**Potential Impact:** 4
**Recommendation:** 3
**Confidence:** 3

**Review:**

The paper presents a full pipeline of a novel data collection device, a data processing pipeline, and a model for learning locomotion affordances. The data collection uses vision tools to reconstruct a 3D radiance field of the environment and human feet movements, therefore allowing the training of navigatable area predictors and robot navigation policies in simulation. The authors also released a new dataset of collected data.

This paper makes a clear contribution to the field of real-world navigation tasks. I can imagine the collected data and the simulator integration will be useful for various tasks.

Regarding the newly proposed affordance learning algorithm, it seems that the key insight here is the label generation process based on DINO features. However, I think more justification is needed here. Although this approach can reduce the amount of false negatives in labels, it also suffers from introducing additional false positives, because such semantic-similarity-based labels may fail to capture "the nuanced human preferences encountered in complex scenarios", such as the example of sidewalks mentioned by the authors in L129.

I agree that this might be a good enough strategy for generating low-cost labels with decent quality, but more visualizations and consensus analysis with human "groundtruth" labels should be included, to demonstrate the effectiveness/trade-offs of this approach.

**Quality Of The Limitations Section:**

3

**Questions For Rebuttal:**

I do not have specific questions. I encourage the authors to provide additional analysis of the semantic similarity-based labeling procedure.

**Robotics Focus:**

4

**Summary Of Paper:**

This paper presents a novel pipeline for locomotion data collection

**Summary Of Recommendation:**

This paper is making solid contributions, but its scientific contribution is rather low, and it could benefit from more analysis of the data.

---

### Decision · Program_Chairs · 2024-09-04

**Decision:**

Accept

**Comment:**

The focus of this paper is on the development of a data collection and simulation pipeline for legged locomotion.

-- The strength of this paper is on the data collection pipeline. Generally the reviewers were favorable about that. One reviewer thought the data collection rig was too complex.

-- There was a question about whether the human footstep affordances collected in the data would generalize well to different robot morphologies.

-- There was also a question about how well DINO v2 features would adequately capture footfall affordances and transfer them to novel scenes (reviewer ZQSK). Perhaps this is something that can be validated experimentally.